# Patterns of Nitrogen and Phosphorus along a Chronosequence of Tea Plantations in Subtropical China

Shun Zou [1,2,*], Chumin Huang [1], Yang Chen [1,2], Xiaolong Bai [1,2], Wangjun Li [1,2] and Bin He [1,2]

[1] School of Ecological Engineering, Guizhou University of Engineering Science, Bijie 551700, China; huangcm@gues.edu.cn (C.H.); chenyang@gues.edu.cn (Y.C.); baixiaolong@gues.edu.cn (X.B.); teesn470@gues.edu.cn (W.L.); hebin@gues.edu.cn (B.H.)

[2] Guizhou Province Key Laboratory of Ecological Protection and Restoration of Typical Plateau Wetlands, Guizhou University of Engineering Science, Bijie 551700, China

[*] Correspondence: zoushun@gues.edu.cn

**Abstract:** Nitrogen (N) and phosphorus (P) play crucial roles in tea planting, but information on how the long-term excessive application of chemical N fertilizer affected N and P in subtropical tea plantations remains limited. In this study, soil and leaf samples were collected along a chronosequence of tea plantations (0-, 5-, 10-, 18- and 23-year-old plantations) with excessive N input but no P application to investigate the effects of planting age on N and P availability. Diverse soil N forms and P fractions, and the concentrations of leaf N and P were measured. The results showed that (1) $NO_3^-$-N and dissolved organic nitrogen (DON) concentrations in both topsoil and subsoil, and the N concentration of mature leaf showed significant upward trends with tea plantation age; (2) the concentrations of available phosphorus (AP), $NaHCO_3$-$P_i$ in labile P pool, NaOH-$P_i$ and D.HCl-$P_i$ in moderately labile P pool in both layers of soil and that the P concentration of mature leaf also increased with age; (3) the N concentration of mature leaves had closely significant logarithmic relations with $NO_3^-$-N concentration, while the TP (total P) concentration of mature leaves had significant positive correlations with AP; and (4) the ratio of N/P in tea leaves indicated a slightly P limitation in tea plantations. We concluded that the $NO_3^-$-N concentration, instead of TN (total N) of $NH_4^+$-N, would be a good indicator to reflect N availability for tea trees, and the increasing of $NO_3^-$-N concentration in soil has a diminishing promoting effect on the TN concentration of mature leaves. The long-term application of chemical N fertilizer had not lead to serious P limitation in subtropical tea plantations. Generally, our study could contribute to improving our understanding of N and P availability and optimizing fertilization management in subtropical tea plantations.

**Keywords:** nitrogen and phosphorus; nitrate; phosphorus fractionation; soil layer; tea plantation age

## 1. Introduction

Nitrogen (N) and phosphorus (P) play crucial roles in agriculture [1]; for instance, N is a major component of proteins, chlorophyll and nucleic acids, which are essential for cell division, photosynthesis and overall plant growth. It enhances vigorous shoot growth and root development [2]. P is vital for the transfer of genetic information and the genetic material formation of DNA and RNA and contributes to plant stress resistance [3]. However, excessive use of fertilizers can lead to the soil nutrients being out of balance and to environmental issues [4]. Therefore, maintaining appropriate N and P concentrations in soil and plants is crucial for sustainable agriculture [5].

The long-term excessive application of chemical fertilizers is a widespread problem in agricultural production [6,7]. From 2009 to 2019, N chemical fertilizer production greatly increased from 98 million tons to 123 million tons the world over [8]. However, less than half of N fertilizers are effective in crop production and synthetic N was the most important contributor to global anthropogenic N entering the environment [8]. Especially in modern tea (*Camellia sinensis* (L.) O. Ktze) plantations under high intensity management, the overuse

of chemical N fertilizers is a serious problem [4]. Buds and tender leaves are economically valuable parts of tea trees. The rapid sprouting of buds and growth of tender leaves is highly dependent on N fertilizers. Therefore, the application of N fertilizer is widely adopted in tea plantations to improve tea yield [9]. According to research, the average amount of N fertilizers applied in tea plantations was about 800 kg N hm$^{-2}$ a year globally [8]. Current levels of N inputs in Chinese tea plantations range from 281 to 745 kg N hm$^{-2}$ every year, with an average of 444 kg N hm$^{-2}$ every year [10].

To apply a rational amount of fertilization, and achieve the sustainable and healthy development goals of tea plantations, the basic requirement lies in understanding how the long-term excessive application of chemical N fertilizers impacts on the status of N and P concentrations in the soil and plants of tea plantations. Soil N exists in various forms, which can be broadly categorized into organic N and inorganic N [11]. Organic N is held within organic matter, such as decomposed plant and animal residues, and most of it is not readily available for plant uptake [12]. $NO_3^-$ and ammonium ($NH_4^+$) are major forms of inorganic N, and can be directly absorbed by plant roots [2]. When N fertilizer is input into soil, it can convert into inorganic N by soil bacteria and be taken up by plants. For example, urea, as a commonly used N fertilizer in agricultural practices, undergoes a series of transformations in the soil, known as urea hydrolysis and nitrification, and converts into available N forms of $NH_4^+$ and $NO_3^-$ [13]. It is important to note that most plants can take up and utilize both $NH_4^+$ and $NO_3^-$ to some extent. However, certain plants have evolved a preference for one form over the other. According to plot experiments, tea tree exhibits a preference for $NH_4^+$, indicating its preference for and ability to uptake and utilize $NH_4^+$ as a primary source of N [2]. On the other hand, excessive N chemical fertilizer entering tea plantation soil may lead to soil P limitation. In particular, tea plantations usually apply a large amount of N fertilizer but little P fertilizer in practice, which may result in low soil P concentration [14,15]. N and P are both essential nutrients for plant growth, and the ratio of N to P in leaves is commonly used to indicate N or P limitation [16].

The purpose of this study was to evaluate the impact of tea plantation age on patterns of N and P in soil and leaves in a chronosequence of tea plantations with excessive application of chemical N fertilizer only and the same managing practices. Here, we hypothesize that (1) the total N concentrations of soil and leaves increased; (2) the total P concentrations of soil and leaves decreased and soil P pools significantly changed with tea plantation age; (3) tea trees preferred $NH_4^+$ and leaf TN concentration had significant correlation with soil $NH_4^+$ concentration; and (4) the ratio of N/P in tea leaves increased with age, indicating P limitation in tea plantations.

## 2. Materials and Methods

### 2.1. Description of the Sample Site

The study was carried out at the tea planting experimental station in MeiTan County, southwestern China. The station is located at 27°20′18″ N, 107°15′36″ E, with an elevation of 778.7 m. It belongs to a subtropical monsoon climate zone, with an annual temperature 14.9 °C (the coldest month is January with 3.8 °C, and the hottest month is July with 25.1 °C) and annual precipitation 1137.1 mm, about 75% of which is concentrated in the spring and summer seasons. The station has time-series tea plantations, including 5a (planted in the year of 2019), 10a (planted in the year of 2013), 18a (planted in the year of 2005) and 23a (planted in the year of 2000); an adjacent pine (*Pinus massoniana* Lamb.) forest was set as a control (0a) with sparse tea trees in the understory. The tea seeds of the pine forest had escaped from tea plantations. These tea plantations had the same planting and management practices, for example, tea picking, pruning and fertilizing. Consistent with local farmers' fertilization habits, the fertilization of tea plantations in the station was three times a year, performed separately in May, July and November, and in total about 1200.0 kg hm$^{-2}$ urea (554.4 kg hm$^{-2}$ N) was evenly sprinkled on tea plantations every year without any other phosphorus fertilizer or organic fertilizer.

## 2.2. Collection of Soil and Leaf Samples

The sampling time was selected in April for the minimal impact of a single fertilization practice within a year. In April 2023, we collected soil and leaf samples in tea plantations with ages of 0a, 5a, 10a, 18a and 23a. Three 5 m × 5 m sample plots were set in each tea plantation. In a sample plot, the top-layer soil (0–20 cm) and sublayer soil (20–40 cm) were drilled using a stainless steel soil drill, and young leaf (except the 0a plot, as it did not have enough leaves) and mature leaf samples were collected. In total, 30 soil samples, 12 young leaf samples and 15 mature leaf samples, were gathered for chemical analysis.

## 2.3. Chemical Analysis of Soil Sample

For each soil sample, TN, nitrate ($NO_3^-$-N), ammonium ($NH_4^+$-N), nitrite ($NO_2^-$-N), dissolved organic nitrogen (DON) concentrations and stable nitrogen isotope ratio ($\delta^{15}N$) were determined. TN concentration was measured using Kjeldahl method [17]. After extracting by KCl solution, $NO_3^-$-N concentration was measured using dual-wavelength colorimetric method and $NH_4^+$-N concentration was measured using indophenol blue spectrophotometric method; $NO_2^-$-N concentration was determined using Griess–Saltzman method [18]. DON could be calculated as the difference between the total dissolved nitrogen (TDN) and the $NO_3^-$-N and $NH_4^+$-N concentrations, and the TDN concentration was measured using the alkaline potassium persulfate digestion–UV spectrophotometric method [19]. $\delta^{15}N$ was measured by using stable isotope mass spectrometer (Stable Isotope Analysis System-Delta plus AD, Thermo-Finnigan, Bremen, Germany) and $\delta^{15}N$ was calculated in thousandth units (‰) according to international standard formulas [20].

At the same time, the TP, available phosphorus (AP) concentrations and soil phosphorus components were determined for every soil sample by modifying the Hedley fractionation method. TP concentration was measured using Mo-Sb anti spectrophotometer method and AP concentration using vanadium molybdate blue colorimetric method [21]. Soil phosphorus components were determined using the Hedley fractionation method modified by Tiessen [22,23]. This method determined three groups of phosphorus fractions by their solubility, including labile P ($H_2O$-$P_i$, $NaHCO_3$-$P_i$ and $NaHCO_3$-$P_o$), moderately labile P ($NaOH$-$P_i$, $NaOH$-$P_o$ and D.HCl-$P_i$ extracted by 1 M HCl) and stable P pools (C.HCl-$P_i$ and C.HCl-$P_o$ extracted by concentrated HCl and Residual-$P_t$).

## 2.4. Chemical Analysis of Leaf Sample

For each leaf sample, TN and TP concentrations were determined. After $H_2SO_4$-$H_2O_2$ digestion, TN concentration of leaf sample was measured using Kjeldahl method and TP concentration was measured using vanadium molybdate blue colorimetric method [24].

## 2.5. Statistical Analysis

Before statistical analysis, the normality and homogeneity of the data were checked using the Shapiro–Wilk test and Levene test, respectively. Data were ln-transformed or square-rooted whenever necessary. A two-way ANOVA was used to test the effects of age (tea plantation age) and soil layer (top layer and sublayer) and their interaction on the concentrations of N and P of soil, and to test the effects of age (tea plantation age) and leaf type (young and mature leaf) and their interaction on the concentrations of N and P of leaf. For relationships between soil N and leaf N, and soil P and leaf P, the Spearman correlation analysis was performed. The linear regression model with the natural logarithmic transformation of variables was used to assess the nonlinear relationship between leaf TN concentration and soil $NO_3^-$-N concentration. Statistical analysis was carried out using IBM SPSS Statistics 27 (IBM Inc., New York, NY, USA) and the figures were plotted by OriginPro 2021 (OriginLab Corporation, Northampton, MA, USA).

## 3. Results

### 3.1. Pattern of Soil Nitrogen Concentration

The concentrations of soil $NO_3^-$-N (Figure 1B), $NH_4^+$-N (Figure 1C), $NO_2^-$-N (Figure 1D), DON (Figure 1E) and $\delta^{15}N$ (Figure 1F) were significantly affected by tea plantation age (Figure 1, $p < 0.001$). In particular, $NO_3^-$-N, DON concentrations and $\delta^{15}N$ in the top layer (0–20 cm) and sublayer (20–40 cm) of the soil, and the TN (Figure 1A) of the sublayer soil had significant ($p < 0.05$) upward trends with tea plantation age. On the other hand, compared to the sublayer, the top-layer soil had higher TN, $NO_3^-$-N, $NH_4^+$-N, $NO_2^-$-N and DON concentrations. Significant interaction effects indicated that $NO_3^-$-N, $NH_4^+$-N and $NO_2^-$-N concentrations had different changes between the top-layer and sublayer soil with tea plantation age.

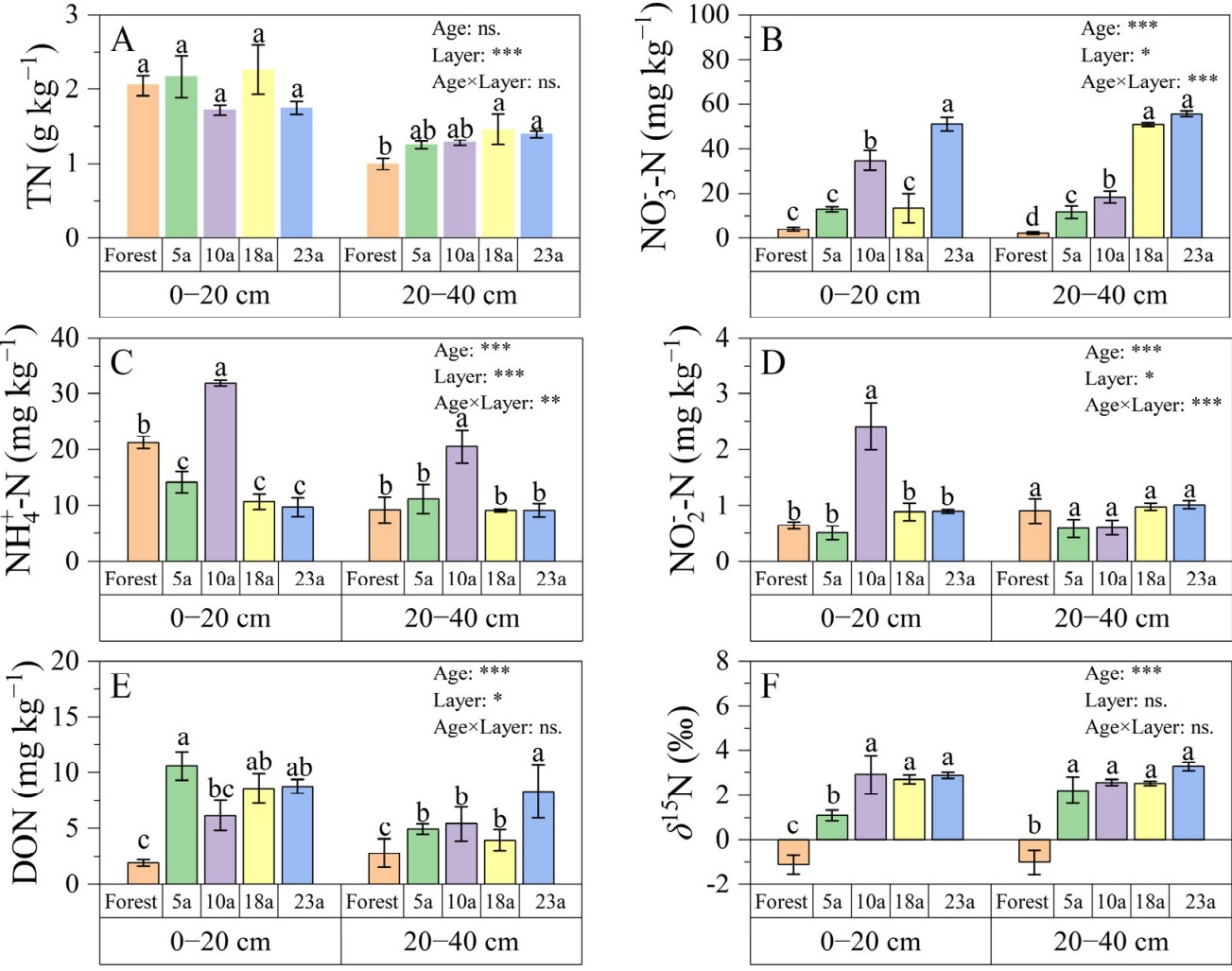

**Figure 1.** Soil nitrogen concentrations of top-layer (0–20 cm) and sublayer (20–40 cm) with planting age in tea plantation. (**A**), total nitrogen (TN); (**B**), nitrate ($NO_3^-$-N); (**C**), ammonium ($NH_4^+$-N); (**D**), nitrite $NO_2^-$-N); (**E**), dissolved organic nitrogen (DON); (**F**), stable nitrogen isotope ratio ($\delta^{15}N$). Different lowercase letters indicate significant differences at $p < 0.05$. Bars indicate means ± standard errors. A two-way ANOVA method was used to test the effects of age (tea plantation age), soil layer (top layer and sublayer) and their interaction on the concentrations of diverse soil N forms and $\delta^{15}N$. *, $0.01 \leq p < 0.05$; **, $0.001 \leq p < 0.01$; ***, $p < 0.001$; ns, not significant.

### 3.2. Pattern of Soil Phosphorus Concentration

The concentrations of soil TP (Figure 2A) and AP (Figure 2B) were significantly affected by tea plantation age, soil layer and their interaction ($p < 0.05$). The TP concentration of top-layer soil and AP concentrations of both layers of soil had significant ($p < 0.05$) upward trends with tea plantation age. Additionally, compared to the sublayer, top-layer soil had higher TP and AP concentrations.

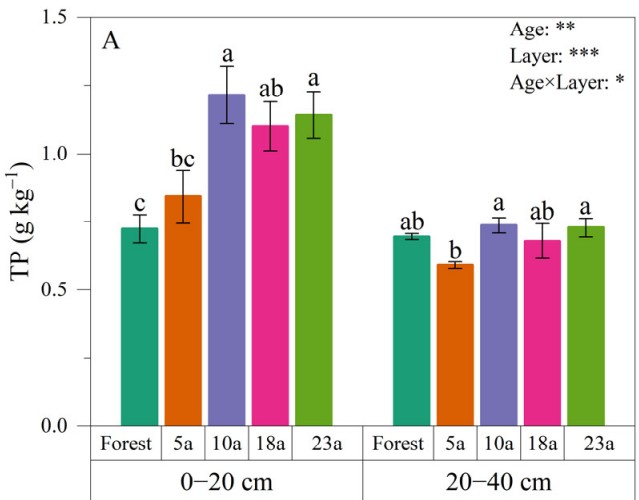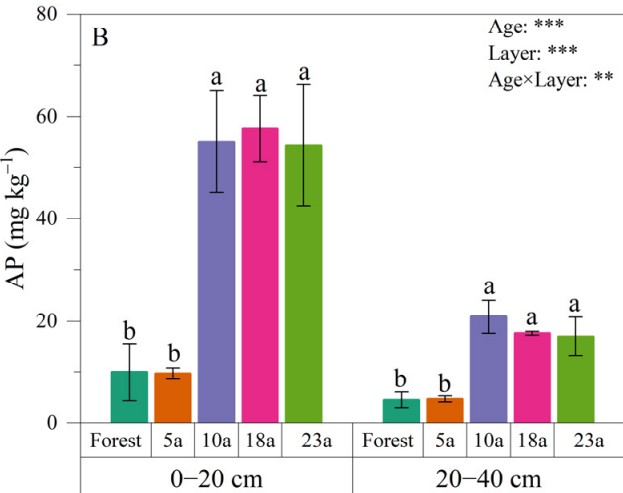

**Figure 2.** Soil total phosphorus (TP, (**A**)) and available phosphorus (AP, (**B**)) concentrations of top layer (0–20 cm) and sublayer (20–40 cm) with planting age in tea plantation. Different lowercase letters indicate significant differences at $p < 0.05$. Bars indicate means ± standard errors. A two-way ANOVA method was used to test the effects of age (tea plantation age), soil layer (top layer and sublayer), and their interaction on the concentrations of soil TP and AP. *, $0.01 \leq p < 0.05$; **, $0.001 \leq p < 0.01$; ***, $p < 0.001$.

Furthermore, the analysis of P components, using Hedley fractionation method modified by Tiessen (Figure 3), indicated that $NaHCO_3$-$P_i$ (Figure 3B) in labile P pool, and $NaOH$-$P_i$ (Figure 3D) and $D.HCl$-$P_i$ (Figure 3E) in moderately labile P pool, had significant ($p < 0.05$) upward trends with tea plantation age in both soil layers, but stable P pool ($C.HCl$-$P_i$, $C.HCl$-$P_o$ and Residual-$P_t$) of both soil layers significantly ($p < 0.05$) decreased with tea plantation age. Finally, except stable P pools, all the other P fractions in topsoil were higher than that in the sublayer soil.

### 3.3. Pattern of Tea Leaf Nitrogen and Phosphorus Concentrations

The concentrations of leaf TN (Figure 4A), TP (Figure 4B) and N/P ratio (Figure 4C) were significantly affected by tea plantation age, leaf type and their interaction ($p < 0.001$). In particular, the concentrations of mature leaf TN and TP showed significant ($p < 0.05$) upward trends with tea plantation age, while the N/P ratio in mature leaves took on a firstly decreasing and then increasing change. Undoubtedly, young leaves had higher TN and TP concentrations but a lower N/P ratio than mature leaves.

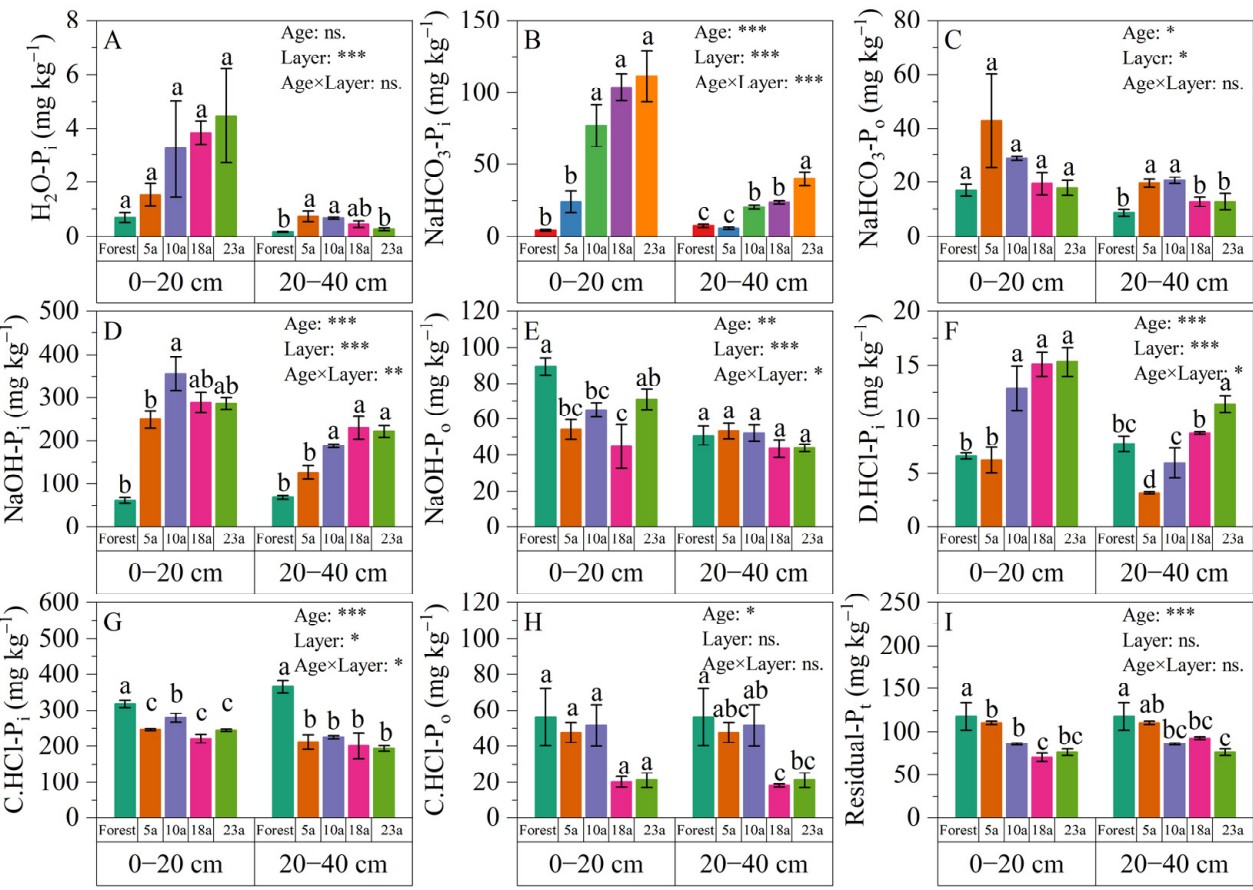

**Figure 3.** Soil phosphorus components by modifying Hedley fractionation method of top layer (0–20 cm) and sublayer (20–40 cm) with planting age in tea plantation. (**A**–**C**), labile P pools ($H_2O$-$P_i$, $NaHCO_3$-$P_i$ and $NaHCO_3$-$P_o$); (**D**–**F**), moderately labile P pools (NaOH-$P_i$, NaOH-$P_o$ and D.HCl-$P_i$ extracted by 1 M HCl); (**G**–**I**), stable P pools (C.HCl-$P_i$ and C.HCl-$P_o$ extracted by concentrated HCl and Residual-$P_t$). Different lowercase letters indicate significant differences at $p < 0.05$. Bars indicate means ± standard errors. A two-way ANOVA method was used to test the effects of age (tea plantation age), soil layer (top layer and sublayer), and their interaction on the concentrations of diverse soil P fractions. *, $0.01 \le p < 0.05$; **, $0.001 \le p < 0.01$; ***, $p < 0.001$; ns, not significant.

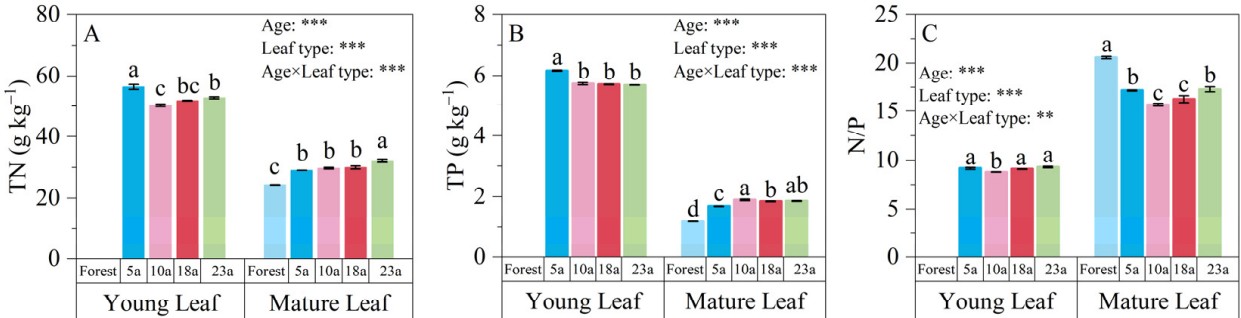

**Figure 4.** Total nitrogen (TN, (**A**)), total phosphorus (TP, (**B**)) and N/P (**C**) concentrations in young leaves and mature leaves of tea with planting age in tea plantation. Different lowercase letters indicate significant differences at $p < 0.05$. Bars indicate means ± standard errors. A two-way ANOVA method was used to test the effects of age (tea plantation age), leaf type (young leaf and mature leaf) and their interaction on the concentrations of leaf N, P and N/P ratio. **, $0.001 \le p < 0.01$; ***, $p < 0.001$.

### 3.4. Correlations between Soil and Leaf Nitrogen Concentrations

According to the results of the correlation analysis (Table 1), we found that the TN concentration of mature leaves had significant positive correlations with $NO_3^-$-N concentrations in both the top-layer ($r = 0.75$, $p < 0.01$) and sublayer ($r = 0.89$, $p < 0.001$) soil, a significant positive correlation with ($NO_2^-$-N concentration in the top-layer ($r = 0.56$, $p = 0.03$) soil and a significant positive correlation with the TN concentration of the top-layer ($r = 0.58$, $p = 0.02$) soil. Otherwise, the TN concentration of mature leaves had significant positive correlations with $\delta^{15}N$ in both the top-layer ($r = 0.77$, $p < 0.01$) and sublayer ($r = 0.72$, $p < 0.001$) soil, which could indicate the net mineralization rate and nitrification rate of soil.

**Table 1.** The Spearman correlation between the nitrogen concentration of tea leaves and the soil nitrogen concentration.

| Soil Layer | | TN of Young Leaf | | | TN of Mature Leaf | | |
|---|---|---|---|---|---|---|---|
| | | *n* | *r* | *p* | *n* | *r* | *p* |
| Topsoil (0–20 cm) | TN | 12 | 0.33 | =0.30 | 15 | −0.36 | =0.19 |
| | $NO_3^-$-N | 12 | −0.23 | =0.47 | 15 | 0.75 | **<0.01** |
| | $NH_4^+$-N | 12 | −0.38 | =0.23 | 15 | −0.39 | =0.15 |
| | $NO_2^-$-N | 12 | −0.76 | **<0.01** | 15 | 0.56 | **=0.03** |
| | DON | 12 | 0.55 | =0.06 | 15 | 0.40 | =0.14 |
| | $\delta^{15}N$ | 12 | −0.45 | =0.14 | 15 | 0.77 | **<0.01** |
| Subsoil (20–40 cm) | TN | 12 | 0.07 | =0.84 | 15 | 0.58 | **=0.02** |
| | $NO_3^-$-N | 12 | −0.01 | =0.98 | 15 | 0.89 | **<0.001** |
| | $NH_4^+$-N | 12 | −0.38 | =0.22 | 15 | 0.09 | =0.75 |
| | $NO_2^-$-N | 12 | 0.03 | =0.93 | 15 | 0.20 | =0.47 |
| | DON | 12 | 0.34 | =0.28 | 15 | 0.44 | =0.10 |
| | $\delta^{15}N$ | 12 | 0.01 | =0.99 | 15 | 0.72 | **<0.01** |

Furthermore, the regression models indicated that the TN of mature leaves had closely significant logarithmic relations with $NO_3^-$-N concentrations in both top-layer ($R^2 = 0.54$, $p < 0.01$) and sublayer ($R^2 = 0.82$, $p < 0.001$) soil (Figure 5).

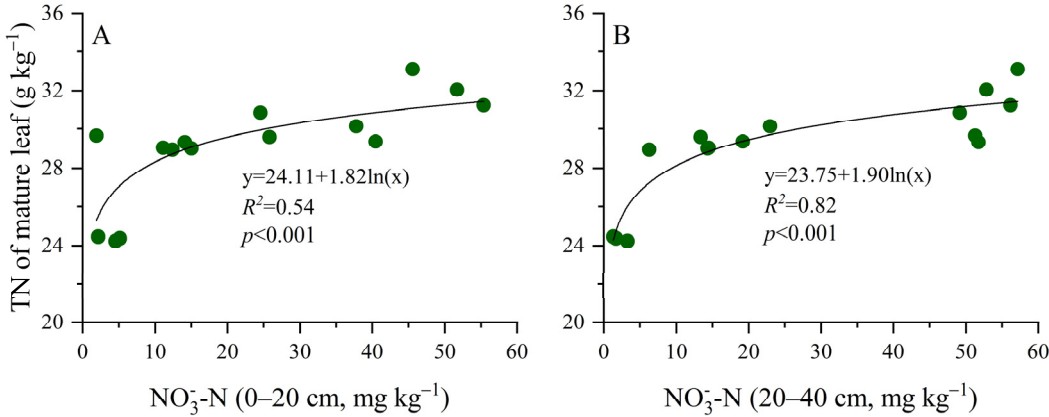

**Figure 5.** The regression models between the nitrogen concentration of tea leaves and soil nitrate ($NO_3^-$-N) concentrations of top-layer (0–20 cm, (**A**)) and sublayer (20–40 cm, (**B**)) soil.

### 3.5. Correlations between Soil and Leaf Phosphorus Concentrations

Based on the results of correlation analysis (Table 2), we found that the TP concentration of mature leaves had significant positive correlations with the TP and AP concentrations in both the top-layer ($r = 0.89$, $p < 0.001$; $r = 0.51$, $p = 0.05$) and sublayer ($r = 0.88$, $p < 0.001$; $r = 0.89$, $p < 0.001$) soil. Further, $NaHCO_3$-$P_i$ in labile P pool and NaOH-$P_i$ in moderately labile P pool in both the top-layer ($r = 0.81$, $p < 0.001$; $r = 0.68$, $p < 0.01$) and

sublayer ($r = 0.82$, $p < 0.001$; $r = 0.81$, $p < 0.001$) soil had significant positive effects on the TP concentrations of mature leaves. In addition, $H_2O$-$P_i$ of labile P ($r = 0.74$, $p < 0.01$) and D.HCl-$P_i$ of moderately labile P ($r = 0.70$, $p < 0.01$) in top-layer soil significantly correlated with the TP concentrations of mature leaves, while Residual-$P_t$ of stable P pool in sublayer soil had a significant negative relationship ($r = -0.74$, $p < 0.01$) with the TP concentrations of mature leaves. None of the organic P fractions had significant effects on the TP concentrations of mature leaves. Conversely, the TP concentrations of young leaves had significant negative relationships with AP, NaHCO$_3$-$P_i$ and D.HCl-$P_i$ in both soil layers and significant negative relationships with NaOH-$P_i$ and Residual-$P_t$ in sublayer soil.

**Table 2.** The Spearman correlation between the phosphorus concentration of tea leaves and the soil phosphorus components.

| Soil Layer | | TP of Young Leaf | | | TP of Mature Leaf | | |
|---|---|---|---|---|---|---|---|
| | | *n* | *r* | *p* | *n* | *r* | *p* |
| Topsoil (0–20 cm) | TP | 12 | −0.49 | =0.11 | 15 | 0.89 | **<0.001** |
| | AP | 12 | −0.67 | **=0.02** | 15 | 0.88 | **<0.001** |
| | H$_2$O-P$_i$ | 12 | −0.38 | =0.22 | 15 | 0.74 | **<0.01** |
| | NaHCO$_3$-P$_i$ | 12 | −0.83 | **<0.01** | 15 | 0.81 | **<0.001** |
| | NaHCO$_3$-P$_o$ | 12 | 0.46 | =0.13 | 15 | 0.19 | =0.51 |
| | NaOH-P$_i$ | 12 | −0.31 | =0.33 | 15 | 0.82 | **<0.001** |
| | NaOH-P$_o$ | 12 | −0.45 | =0.14 | 15 | −0.20 | =0.48 |
| | D.HCl-P$_i$ | 12 | −0.71 | **=0.01** | 15 | 0.70 | **<0.01** |
| | C.HCl-P$_i$ | 12 | 0.13 | =0.68 | 15 | −0.22 | =0.42 |
| | C.HCl-P$_o$ | 12 | 0.30 | =0.34 | 15 | −0.45 | =0.09 |
| | Residual-P$_t$ | 12 | 0.40 | =0.20 | 15 | −0.45 | =0.09 |
| Subsoil (20–40 cm) | TP | 12 | −0.47 | =0.12 | 15 | 0.51 | **=0.05** |
| | AP | 12 | −0.74 | **<0.01** | 15 | 0.89 | **<0.001** |
| | H$_2$O-P$_i$ | 12 | 0.49 | =0.11 | 15 | 0.44 | =0.10 |
| | NaHCO$_3$-P$_i$ | 12 | −0.66 | **=0.02** | 15 | 0.68 | **<0.01** |
| | NaHCO$_3$-P$_o$ | 12 | 0.36 | =0.26 | 15 | 0.48 | =0.07 |
| | NaOH-P$_i$ | 12 | −0.74 | **<0.01** | 15 | 0.81 | **<0.001** |
| | NaOH-P$_o$ | 12 | 0.36 | =0.24 | 15 | −0.11 | =0.68 |
| | D.HCl-P$_i$ | 12 | −0.61 | **=0.04** | 15 | 0.25 | =0.38 |
| | C.HCl-P$_i$ | 12 | 0.22 | =0.48 | 15 | −0.35 | =0.20 |
| | C.HCl-P$_o$ | 12 | 0.48 | =0.12 | 15 | −0.24 | =0.39 |
| | Residual-P$_t$ | 12 | 0.59 | **=0.04** | 15 | −0.74 | **<0.01** |

## 4. Discussion

### 4.1. Patterns of Nitrogen in Soil and Leaves

The excessive application of chemical N fertilizer or N addition may greatly improve the TN in soil [21,25,26]. However, in our results, the TN concentration in top-layer soil did not change with plantation age, and the soil TN concentration of tea plantations even showed no significant difference with that of adjacent forest soil with no application of N fertilizer. The TN concentration in the sublayer soil had a significant but small change. It indicated that the N input in tea plantations was not mainly stored in soil, and that TN concentration may not be an adequate indicator of the N input of tea plantations. Despite storing a part of N in tea trees, much of the N input might enter the environment [5,27]. The same goes for the $NH_4^+$-N and $NO_2^-$-N concentration. The $NO_3^-$-N and DON concentrations in both the soil layers showed significant increasing trends with the tea plantation age [28], which typical for $NO_3^-$-N, indicating that decades of continuous N fertilizer application significantly increase $NO_3^-$-N concentration. $\delta^{15}N$ could synthetically represent the net mineralization and nitrification rate of soil. Increasing the $\delta^{15}N$ value reflected the higher net mineralization and nitrification rate of soil with tea plantation age [29,30]. More N fertilizer and organic N had been converted into $NO_3^-$-N, which led to a higher soil $NO_3^-$-N concentration in old

tea plantations. In general, top-layer soil had a higher N concentration than sublayer soil, except for $NO_3^-$-N, possibly because of its readiness to leach into the soil [5,21].

Meanwhile, the TN concentrations of mature leaves increased with the chronosequence of tea plantations, but those of young leaves did not exhibit a significant trend. Consistent with previous research, young leaves had higher TN concentrations than mature leaves [31]. Young leaves are actively engaged in photosynthesis and growth, which require higher levels of N to support the synthesis of chlorophyll and other proteins involved in photosynthesis [32].

### 4.2. Patterns of Phosphorus in Soil and Leaves

In our hypothesis (1), total P would decrease with no application of P fertilizer [14,15]. On the contrary, the TP concentration in topsoil had an upward trend, especially in older tea plantations (10a, 18a and 23a). Plants taking up P from deep soil and leaf litter decomposition may contribute to the accumulation of P in the topsoil [21]. This could also explain why P concentrations of most fractions were higher in the topsoil than subsoil. AP concentration increased with tea plantation age, which may be related to the rising of the $NaHCO_3$-$P_i$ concentration in labile P pool, and the $NaOH$-$P_i$ and $D.HCl$-$P_i$ concentrations in moderately labile P pool [22,23]. At the same time, stable P pool ($C.HCl$-$P_i$, $C.HCl$-$P_o$ and Residual-$P_t$) significant decreased with tea plantation age. This might indicate that P forms were internally transforming, and stable P pool was activated to supplement moderately labile and labile P pools.

On the other hand, the TP concentration of mature leaves increased with age [21], but that of young leaves in older tea plantations (10a, 18a and 23a) was lower than in 5-year-old plantations. We hypothesized that the ratio of N/P in tea leaves would increase with age and with P limitation. However, tea tree maintained its N/P ratio in young leaves, and it even slightly decreased in mature leaves. Generally, when the N/P ratio of mature leaves is higher than 16, plant growth is limited by P [24,33]. Although tea trees in all plantations were slightly limited by P, tea trees could commendably absorb P [21].

### 4.3. Soil Controls Leaf Nitrogen and Phosphorus Concentrations

The plant uptake of N and P from soil, and the soil availability of N and P, closely affect leaf N and P concentrations [34,35]. In this study, we found that the TN concentration of mature leaves closely correlated with $NO_3^-$-N concentration in soil. Furthermore, the regression model between $NO_3^-$-N concentration in the subsoil and the TN concentration of mature leaves was significant ($R^2 = 0.82$, $p < 0.001$) (logarithmic model (y = 23.75 + 1.90ln(x))). This indicates that an increase in $NO_3^-$-N concentration in soil has a diminishing promoting effect on the TN concentration of mature leaves. At the same time, $NO_3^-$ ions can be readily leached from the soil and contaminate groundwater or nearby surface water bodies [36]. Therefore, considering $NO_3^-$-N utilization efficiency by mature leaves and environmental impacts, 25 mg kg$^{-1}$ $NO_3^-$-N in soil may be a sufficient concentration. According to research, tea trees prefer to absorb $NH_4^+$-N over $NO_3^-$-N as a primary source of N, due to the weak nitrate reductase activity of tea trees in reducing $NO_3^-$-N to $NH_4^+$-N [2]. However, our results do not support this. The absorbing of $NH_4^+$-N might be hindered by the extremely low soil pH values in these tea plantations [4,21]. In subtropical tea plantations, the $NO_3^-$-N concentration, instead of TN of $NH_4^+$-N, would be a good indicator reflecting N availability for tea trees.

On the other hand, we found that soil P concentration was closely correlated with mature leaf P concentration, but different P forms had distinct effects [37,38]. Organic P is present in the soil as part of organic matter, such as decomposed plant and animal residues. This form of P is not directly available to plants [39]. The organic P pool mainly includes $NaHCO_3$-$P_o$, $NaOH$-$P_o$ and $C.HCl$-$P_o$ according to the Hedley fractionation method as modified by Tiessen [22,23]. None of these forms of organic P had a significant effect on leaf P concentration, indicating that the organic P pool had extremely low availability for tea trees [39]. The soil TP showed a close correlation with mature leaf P concentration.

The high P concentration litter could explain the massive TP enrichment in the topsoil, instead of the opposite [21]. Therefore, the TP concentration is not sufficient to represent the P status of tea plantations [40]. Meanwhile, soil AP could be a good indicator of P availability, because of its close relationship with mature leaf P concentration. Furthermore, $NaHCO_3$-$P_i$ in labile P pool and $NaOH$-$P_i$ in moderately labile P pool might be excellent available P fractions for tea trees [22,23].

## 5. Conclusions

According to our results and analysis, we found that the TN did not significantly accumulate in soil, especially in the top-layer soil, but the TN concentrations of mature leaves increased with the chronosequence of tea plantations. The $NO_3^-$-N concentration, instead of the TN of $NH_4^+$-N, would be a good indicator reflecting N availability for tea trees. In soil, 25 mg $kg^{-1}$ $NO_3^-$-N may be a sufficient concentration for tea trees. On the other hand, the excessive application of chemical N fertilizer with little P fertilizer did not lead to serious P limitation. The TP concentration in topsoil and mature leaves showed an upward trend because plants possibly take up P from deep soil. The ratio of N/P in tea leaf indicates a slight P limitation in tea plantations. $NaHCO_3$-$P_i$ and $NaOH$-$P_i$ could be excellent available P fractions for tea trees. Our results will be valuable for improving our understanding of the status of N and P in soil and leaves in subtropical tea plantations, which could be conducive to optimizing fertilization management in these tea plantations.

**Author Contributions:** Conceptualization, S.Z.; data curation, C.H.; formal analysis, Y.C. and C.H.; investigation, X.B. and W.L.; methodology, B.H.; project administration, S.Z.; supervision, S.Z.; visualization, S.Z., C.H. and Y.C.; writing—original draft, S.Z., C.H. and X.B.; writing—review and editing, W.L.; and B.H. All authors have read and agreed to the published version of the manuscript.

**Funding:** This work was funded by the Guizhou Provincial Science and Technology Project (qiankehejichu-ZK-[2022]yiban167), the Regional First-class Discipline of Ecology in Guizhou Province (XKTJ[2020]22), the Bijie Talent Team of Biological Protection and Ecological Restoration in Liuchong River Basin (202112) and the Bijie Science and Technology Project (bikelianhe[2023]10).

**Institutional Review Board Statement:** Not applicable.

**Data Availability Statement:** For additional information, please contact the authors.

**Acknowledgments:** We thank the tea planting experimental station for their help in sample collection and processing.

**Conflicts of Interest:** The authors declare no conflicts of interest.

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
