# Peer review of "Patterns of Nitrogen and Phosphorus along a Chronosequence of Tea Plantations in Subtropical China"

_agriculture, doi:10.3390/agriculture14010110_

Round 1
Reviewer 1 Report
Comments and Suggestions for Authors
Review of Agriculture MDPI Special Issue Advances In Nutrient Management in Soil-Plant Systems submission 2807985 Patterns of nitrogen and phosphorus along a chronosequence of tea plantations in subtropical China
An interesting paper that provides new information on amounts and forms of N and P in soils and plants along a tea plantation chronosequence. The results are deserving of publication and data is well presented. Main criticism is some lack of rigor in interpretation. The paper could also benefit from a go-over to address some general English language style and word usage issues (e.g. ‘ulteriorly’ in line 167) but in general is easy to follow and understand.
Specific Comments
Some results need more thorough investigation and explanations, conclusions that are better supported. For example significant large increases in soil total and labile available P in soil are shown with increased age of the stand, despite no external additions of P as fertilizer, manure and continued removal of P in tea harvest. May include somewhere in lines 258 to 269 an estimate of how much N and P are removed in tea harvest from these plantations each year. The increase in soil total P is unusual and difficult to explain with apparent crop removal and no replenishment. The authors attribute this to removal of P from depth by roots and transfer to surface, but no evidence is reported for this. Other possibilities such as mineralization of organic P are suggested but not rigorously considered through use of approaches like P balances. There appears to be decline in stable P forms in soil over time that suggests possible transfer to labile forms. Is there a mechanism for solubilization, mobilization of these P reserves? The implication that tea trees are limited by low P availability does not seem to be supported by the soil data. It is suggested that tea plantation soils are acidic but no pH or background properties for the soils of the plantation treatments is provided. Perhaps the addition of the N fertilizer is acidifying the soil which may affect P solubility depending on the initial pH of the soil. However in apparently acid soils such as these the acidity would tend to reduce rather than increase proportion of soil P held in labile forms.
The soil N data generally indicate that too much N is being applied as evidenced by the accumulation of nitrate evident in older plantations, especially at depth 20-40cm as shown in Fig 1 B. The naturally occurring stable N isotope variations are reported to help support mineralization of organic N as an input to ammonium, but this link is rather tenuous. In lines 286-291, the conclusion to measure soil nitrate rather than ammonium as indicator of N availability is not really justified. I think that both would be appropriate to measure. It seems that nitrate is a good indicator of residual unused nitrogen that has accumulated in soil because of excessive application of N fertilizer beyond tea plant requirement and uptake over the years. However, tea plants may indeed be absorbing and preferentially using ammonium N as they grow but it is the residual nitrate that most strongly reflects the application rate and overall available N status, particularly as it relates to over-application.
Comments on the Quality of English LanguageWould benefit from a go-over to address a few word usage and style issues, but generally easy to follow and understand.
Reviewer 2 Report
Comments and Suggestions for Authors
Patterns of nitrogen and phosphorus along a chronosequence 2 of tea plantations in subtropical China
It is a study with relevant scientific content and framed within the scope of the Agriculture journal.
It is very well written and the methodologies are consistent, which shows the team's scientific experience. However, I will suggest to the authors small changes to the texts that, even if they are of small importance, will help to improve the manuscript a little.
The authors do not respect important abbreviation rules. They use abbreviations without first spelling out what is not correct. See for example TP (line 19), TN (line 22), …. On the other hand, once the abbreviation is made, they should not be used in full again. For example, N and P (line 73, 95, 106-115, …..)….
The authors often used the word content, when they should have used concentration (g/kg, mg/kg, ….). (lines 15, 18, ……)
In the abstract the reader cannot understand the meaning of 0a, 5a, 10a, 18a, 23a. The authors have to find a way to make the reader understand that this means a significant amount of time in years.
